# Machine Learning-Based Rapid Post-Earthquake Damage Detection of RC Resisting-Moment Frame Buildings

**DOI:** 10.3390/s23104694

**Published:** 2023-05-12

**Authors:** Edisson Alberto Moscoso Alcantara, Taiki Saito

**Affiliations:** Department of Architecture and Civil Engineering, Toyohashi University of Technology, Toyohashi 441-8580, Japan; moscoso.alcantara.edisson.alberto.eh@tut.jp

**Keywords:** damage detection, machine learning, intensity measures

## Abstract

This study proposes a methodology to predict the damage condition of Reinforced Concrete (RC) resisting-moment frame buildings using Machine Learning (ML) methods. Structural members of six hundred RC buildings with varying stories and spans in X and Y directions were designed using the virtual work method. Sixty thousand time-history analyses using ten spectrum-matched earthquake records and ten scaling factors were carried out to cover the structures’ elastic and inelastic behavior. The buildings and earthquake records were split randomly into training data and testing data to predict the damage condition of new ones. In order to reduce bias, the random selection of buildings and earthquake records was carried out several times, and the mean and standard deviation of the accuracy were obtained. Moreover, 27 Intensity Measures (IM) based on acceleration, velocity, or displacement from the ground and roof sensor responses were used to capture the building’s behavior features. The ML methods used IMs, the number of stories, and the number of spans in X and Y directions as input data and the maximum inter-story drift ratio as output data. Finally, seven Machine Learning (ML) methods were trained to predict the damage condition of buildings, finding the best set of training buildings, IMs, and ML methods for the highest prediction accuracy.

## 1. Introduction

Various methods for estimating the lateral strength of buildings using sensor information have been proposed. For example, Quispe et al. [1] obtained the capacity curve and the inter-story drift ratios of the Edgardo Rebagliati Martins hospital in Peru using a sparse number of sensors, the wavelet transform method, and the spline shape function. Additionally, Schanze et al. [2] compared the effect of different underground story modeling approaches on Chile’s instrumented Alcazar building office. For this reason, the implementation of instrumented buildings has been increasing recently [3,4,5].

Machine Learning (ML) methods are currently used to predict structures’ damage states [6,7]. For example, Cardellicchio et al. proposed a methodology to interpret defect detection results using Class Activation Maps and Explainable Artificial Intelligence techniques applied to Reinforced Concrete (RC) bridges in Southern Italy [8]. Likewise, the damage condition of buildings after an earthquake is obtained for instrumented buildings [9,10,11]. For instance, Yongjia et al. used IMs to train ML models to classify the damage state of buildings [12]. Additionally, Sajedi et al. proposed a framework for building damage diagnosis using the Support Vector Machines method for damage classification and the Bayesian method for the optimization of input features and hyperparameters. Furthermore, the authors proposed a methodology to predict the damage condition of buildings using the convolutional neural network method using wavelet spectra as input images [13]. This study was improved for three-dimensional structures using the wavelet power spectra and by proposing a methodology to select records in order to increase prediction accuracy. It was applied to two instrumented buildings in Japan [14].

On the other hand, it is possible to predict the damage condition of buildings using information from other buildings with similar features. For example, the fundamental vibration period of buildings, which is obtained approximately using the number of stories or their height [15], is one of the most used parameters to estimate the global stiffness of a structure. Likewise, the lateral load-resisting system can be estimated from the plastic deformation mechanism of the structure [16,17]. On the other hand, the structural acceleration, velocity, or displacement response of a building is influenced by its structural properties and earthquake input characteristics [18]. For example, intensity measures (IM) based on the earthquake ground acceleration have the greatest impact on short-period structures [19]. Therefore, it is possible to establish an archetype (parametric model) of the buildings built by their main structural characteristics and select the best and minimum set of buildings to be the reference to accurately predict the damage condition of the rest of the buildings under various intensity measures. This process is possible using ML methods which capture the main features of input data in order to predict particular output data [20,21].

This study proposes ML methods to predict the damage conditions of RC resisting-moment frame buildings based on the building configurations and IMs of input earthquakes and roof sensor acceleration responses. Incremental Dynamic Analyses (IDA) were carried out to cover the elastic and inelastic behavior of the building. The ML models were trained using a different number of stories and spans in X and Y directions to predict the damage condition of the building expressed by inter-story drift ratios. The proposed ML model can be used to detect post-earthquake damage in many other buildings without sensors, based on earthquake acceleration data observed by sensors installed on the ground or on the roof of the building.

This paper contains five sections: Section 2 presents the methodology and provides an overview of the proposed research procedure. Section 3 shows the information used to obtain the structural responses from the design of the archetype buildings using the virtual work method, selections of records, and the IDAs presenting several intensity measures. Section 4 presents the ML methodology to predict the damage condition of buildings. Finally, Section 5 presents a summary of the conclusions and a discussion of the research results.

## 2. Methodology

Figure 1 shows the procedure employed to obtain the structural responses used in the ML methods. An archetype of the buildings was developed, designed using the virtual work method, and verified using nonlinear static analysis. Additionally, ten records were selected using the uniform hazard spectrum and used as input ground accelerations for Incremental Dynamic Analyses. The result was 60,000 structural linear and nonlinear responses.

The ML methods used IMs, the number of stories, and the number of spans in X and Y directions as the input data and the maximum inter-story drift ratio as the output data. As shown in Figure 2, the ML methods had a training and testing process. The input and output data were used to train the ML model in the training process. The trained ML model was used to obtain new predictions in the testing process, and its accuracy was evaluated using reference results.

The ground motion records were split randomly to obtain the data in the training process (80% of records) and the testing process (10% of records). Likewise, the buildings were split randomly to obtain the data in the training process (10% of buildings) and the testing data (90% of buildings). In order to reduce the bias due to random splitting, this process was carried out several times.

## 3. Structural Design and Input Ground Motions

### 3.1. Archetype of Buildings

#### 3.1.1. Structural Distribution

An RC moment-resisting frame system was used for the archetype of the buildings. Figure 3 shows the plan and elevation view of the archetype of buildings.

Table 1 shows the key design variables of the archetype of buildings. Six hundred buildings were generated.

#### 3.1.2. Structural Design Criterion

The virtual work method was used to determine the lateral strength and design of the members of the buildings. According to the principle of virtual work, a plastic mechanism of the building was assumed under horizontal seismic forces, as shown in Figure 4. From the principle of energy conservation, the external work was equal to the internal work, as defined by Equation (1) [22]:(1)WE=WI
where WE is the external work by the external forces, and WI is the internal work by the internal force of the structural members. Using this formula, the member capacity of the building can be determined from the horizontal force corresponding to the required base shear. The following assumptions were considered in order to obtain the rebar detailing of beams and columns, as shown in Figure 4:-The cross-section of columns is square.-Only plastic hinges are at the base of the columns.-Plastic hinges at the ends of all beams except roof beams.-The beam and column rotations are equal to the roof drift ratio.-The yielding moment of the columns is 1.5 times the yielding moment of the beams.

From the assumptions, the internal and external work were defined by Equations (2) and (3):(2)$$WI=\sum My_{column}\cdot \theta +\sum My_{beam}\cdot \theta$$
(3)$$WE=\sum P_{i}\cdot \delta_{i}$$
where $My_{column}$ is the yield moment of the column, $My_{beam}$ is the yield moment of the beam, θ is the yield rotation of the members, $P_{i}$ is the external load (triangular distribution), and $\delta_{i}$ is the absolute displacement of the building (triangular distribution).

The base shear coefficient used in this criterion was 0.3, based on the Japanese design standard [23]. Then, the base shear force was defined by Equation (4):(4)$$V_{base}=0.3\cdot W_{total}=\sum P_{i}=\frac{n\cdot \left(n+1\right)}{2}\cdot P$$
where $V_{base}$ is the base shear force, $W_{total}$ is the total weight of the building, n is the number of stories of the building, and P is the external force of the first floor, which is obtained from Equation (4). For this plastic mechanism and assuming a triangular deformation distribution, θ is equal to the roof drift ratio, as Equation (5) shows:(5)$$\theta =\frac{\delta_{roof}}{H}=\frac{n\cdot \delta }{H}$$
where $\delta_{roof}$ is the absolute displacement of the roof level, δ is the displacement of the first floor, and H is the total height.

The cross-section size of the columns, in order to avoid axial and shear failure, was obtained by Equation (6):(6)B2≥N0.3·fc and B≤hc3
where B is the size of the column, N is the axial force at the column, $f_{c}$ is the compressive strength of the concrete, and $h_{c}$ is the clear height of the column. The cross-section height range of the beams in order to avoid shear failure was obtained by Equation (7):(7)$$h_{b}=\lfloor \frac{L}{12};\frac{L}{10}\rfloor \;and\;b=\frac{h_{b}}{2}$$
where L is the length of the beam, and its width is b. The yield moment of the column was calculated by Equation (8) [24]:(8)$$My_{column}=\left\{\begin{array}{c} 0.8\cdot a_{tc}\cdot f_{yc}\cdot B+0.5\cdot N\cdot B\cdot \left(1-\frac{N}{B^{2}\cdot f_{c}}\right),\;if\;0<N\leq N_{b}\\ \left(0.8\cdot a_{tc}\cdot f_{yc}\cdot B+0.12\cdot B^{3}\cdot f_{c}\right)\cdot \left(\frac{N_{max}-N}{N_{max}-N_{b}}\right),\;if\;N_{b}<N\leq N_{max} \end{array}\right.$$
where $a_{tc}$ is the rebar area in the tension side of the column section, $f_{yc}$ is the steel-yielding strength used for the column, and $N_{b}$ and $N_{max}$ are the balance and maximum axial force, respectively, which can be approximated by Equations (9) and (10):(9)$$N_{b}=0.4\cdot B^{2}\cdot f_{c}$$
(10)$$N_{max}=B^{2}\cdot \left(f_{c}+\frac{f_{y}}{1.2}\right)$$

The yield moment of the beam was calculated by Equation (11) [24]:(11)$$My_{beam}=0.9\cdot a_{tb}\cdot f_{yb}\cdot \left(h_{b}-r\right)$$
where $a_{tb}$ is the rebar area in the tension side of the beam section, $f_{yb}$ is the steel-yielding strength used for the beam, $h_{b}$ is the height of the beam, and r is the minimum distance of the center of tension rebars to the external fiber of the beam.

#### 3.1.3. Structural Model of the Members

The buildings were modeled as three-dimensional frame structures using the following considerations [14]:For the beams, nonlinear flexural springs are used at both ends of the member. The degrading trilinear slip and bilinear hysteretic models are considered, see Figure 5.For the columns, nonlinear multi-spring cross-section models are used at both ends of the member in order to consider the bidirectional-flexural and axial effects. Bilinear hysteretic models are considered for steel (tension and compression) and concrete (only compression), see Figure 6.Nonlinear shear springs are used in the middle of the beams and columns, and the origin-oriented poly-linear hysteretic model is considered.

The software STERA_3D [25], developed by one of the authors, was used for nonlinear static and dynamic analyses. The structural analysis computation time was optimized by running 16 models in parallel. However, it is possible to make equivalent models in order to optimize the number of parameters and decrease the consumed computation time [26].

#### 3.1.4. Verification of the Structural Design

The structural design of buildings was verified by the nonlinear static analysis (pushover), comparing the base shear force coefficient at the inter-story drift greater or equal to 1/100 with the minimum value of 0.3. Note that the member sizes, rebar distribution, and the minimum and maximum rebar ratio satisfied the recommendations of the Architectural Institute of Japan Standard [24]. Figure 7 shows the box plot of the base shear coefficient of 600 buildings by stories for the inter-story drift of 1/150, 1/100, 1/75, and 1/50. Almost all buildings had a base shear force coefficient of more than 0.3 when the inter-story drift exceeded 1/75.

Table 2 shows the member sizes of columns and beams of each story.

### 3.2. Ground Motion Records

#### 3.2.1. Target Response Spectrum

The Uniform Hazard Spectrum (UHS) is a response spectrum with an equal probability of exceedance of a particular hazard in all structural periods. This paper used Nagoya’s 2500-year return period Uniform Hazard Spectrum (UHS) [23] as the target response spectrum of input earthquakes. This UHS is the acceleration response spectrum of a 5% damping ratio on reference ground (the shear wave velocity in the first 30 m of soil is 292 m/s), corresponding to an exceedance probability of 2% in 50 years.

#### 3.2.2. Records Selection Criterion

Moscoso et al. [14] created a database of 183 records from the ground motion records obtained in the Center of Engineering Strong Motion Data by the USGS and the California Geological Survey [27]. It consists of records with fewer than 3000 samples and PGA greater than 400 gals. Additionally, the record data were cut off between 5% and 95% of the Aries Intensity, where the main energy was released in this time range [28].

Then, the records were further selected in order to obtain the minimum Mean Squared Error (MSE) defined by Equation (12) against the target acceleration response spectra:(12)$$MSE=\frac{1}{N}\cdot \overset{N}{\underset{i=1}{\sum }}{\left(SF_{1}\cdot Sa_{rec}-Sa_{target}\right)}^{2}$$
where $SF_{1}$ is the scaling factor in obtaining the minimum MSE for the evaluated record, $Sa_{rec}$ is the unscaled response spectrum of the evaluated record, and $Sa_{target}$ is the target response spectrum.

Finally, a set of 10 ground motion records and their scaling factors were selected, as shown in Table 3. However, it is necessary to increase the number of records in future studies in order to cover more earthquake features.

Figure 8 shows the target spectrum, the spectrum of the selected records, and the fundamental period range of studied buildings (between 0.237 and 0.609 s).

### 3.3. Incremental Dynamic Analysis

IDA was used to obtain buildings’ linear and nonlinear responses. IDA requires performing a series of nonlinear time-history analyses in which the scale factors ($SF_{2}$) of ground motions are gradually increased until the collapse capacity of the structure is reached [29]. $SF_{2}$ are applied to the records after matching to the target spectrum. The $SF_{2}$ were from 0.10 to 0.30 in increments of 0.10 and from 0.50 to 2.00 in increments of 0.25 in this study.

The IDA curve represents the relationship between Intensity Measure (IM) and Damage Measure (DM). The IMs can be obtained based on either acceleration (A), velocity (V), displacement (D), or by combining them (H: hybrid IM). In this study, 27 IMs were selected, as shown in Table 4. The maximum inter-story drift ratio (story drift) was selected as the DM.

Ten scaling factors of input ground motions were selected for all buildings to capture the linear and nonlinear behavior. As an example, the IDA curve of the three-story (Ns) building with two spans in the x-direction (Nx) and five spans in the y-direction (Ny) is shown in Figure 9.

## 4. ML Methodology to Predict the Damage Condition of the Building

### 4.1. Damage Condition State

The scaling factors of ground motions were determined with reference to the damage conditions in Table 5 so that the story drift could range from No Damage to Collapse condition [13,14].

### 4.2. Input and Output Data for the ML Models

The input data were Intensity Measures (from the ground and/or roof response acceleration), the number of spans in X and Y directions (Nx and Ny), and the number of stories (Ns). The output data were the story drifts.

### 4.3. Case Studies with Different Input Data

In determining the IM from the sensor record, two cases were considered in this study. The record was obtained from the ground motion sensor in the first case, as shown in Figure 10.

In the second case, the records were taken from the ground and roof sensor and IMs were calculated by both records, as shown in Figure 11.

### 4.4. Random Selection of Records and Buildings for the ML Models

In total, 80% and 20% of the records were used for the training and testing processes, respectively. Random record selections were carried out ten times in order to reduce bias. The ML model was trained and tested for each set of records. The accuracy of the ML prediction was evaluated using the coefficient of determination (R^2^), and its mean (R^2^_mean) and standard deviation (in order to evaluate the dispersion) of R^2^ values came from the ten selections. Figure 12 shows the procedure of the record selections of the ML model.

On the other hand, this study randomly selected 10% and 90% of the 600 buildings for training and testing processes, respectively, as shown in Figure 13.

Random building selections were carried out 200 times to reduce bias. The ML model was trained and tested for each set of buildings. The maximum R^2^_mean and its standard deviation of the iterations determined the best training buildings. Figure 14 shows the procedure of the building selection of the ML model.

### 4.5. Machine Learning Methods

The following seven ML methods were used, and their parameters were calibrated after several runs (training process) in order to optimize the prediction. The optimum IMs were obtained from the feature importance level (from 0 to 1), which was obtained using the Gini importance technique [48,49] of the regression tree methods (not for Linear regression or Multilayer perception).

**Linear Regression.** This method assumes that the output (prediction) is linearly dependent on the features. The coefficients (weights) are updated in order to minimize the prediction error obtained from the reference and predicted values [20,50].

**Decision Tree.** This method builds the best decision-making tree by splitting and selecting the order of the roots and leaves. The leaves are chosen when it is not possible for more optimization below those nodes [51,52]. The parameters used in this study are shown in Table 6.

**Random Forest.** This method builds several decision trees (forest) from bootstrapped datasets (a new random dataset with the same size as the original one), increasing its accuracy in this way. The new data to predict are evaluated in the forest [48,53]. The parameters used in this study are shown in Table 7.

**Gradient Boosting (Gradient Boost).** This method makes a tree to obtain residuals instead of predictions. Then, a new predictor is built using the previous predictor (the first one predicts the same value for all and then is updated) and adds the residuals predictor (a learning rate scales it). Therefore, the new predictor is based on the previous tree’s errors [48]. The parameters used in this study are shown in Table 8.

**AdaBoost.** This method fits a regressor onto the original dataset. Then, it fits additional copies of the regressor onto the same dataset, but the weights of instances are adjusted according to the current prediction error [48]. The parameters used in this study are shown in Table 9.

**Extreme Gradient Boosting (XGBoost).** This method is called extreme because it is built with several parts. Like Gradient Boost, the regression tree is obtained using residuals instead of predictions from the similarities and gain values method for splitting and obtaining the thresholds. The pruning method is used to reduce this tree. Additionally, this method uses the regularization parameter to minimize the prediction’s sensitivity to individual observations. Finally, it uses the original previous predictor and learning rate to obtain a new predictor [54]. The parameters used in this study are shown in Table 10.

**Multilayer Perceptron.** This interconnects a group of perceptrons and transmits data to others inspired by the biological neural networks that constitute animal brains. Each connection has weights adjusted to reduce the error [51,55]. The parameters used in this study are shown in Table 11.

### 4.6. Case Study Results

#### 4.6.1. First Case Study Using the Ground Sensor Data

Table 12 shows the ML results for the first case, where the IMs are ordered descending from left to right (collected from the feature importance levels greater than 0.05). Even though the difference between R^2^ and the standard deviation for all the ML methods was generally insignificant, the main results are as follows:The maximum R^2^ obtained by the Random Forest method is 0.942: A95, IMcr, AI, and Ic.The maximum R^2^_mean obtained by the Gradient Boost method is 0.870: A95, AI, Ic, and Imcr.The minimum standard deviation obtained by the Decision Tree method is 0.047 where the main Ims are based on acceleration: A95 and AI.The IM present in all the ML methods is A95.

Figure 15 shows the results of the Random Forest method of the first case. Figure 15a compares the predicted and reference story drift for the maximum R^2^, which was 0.942. Figure 15b shows the normal distribution function of the R^2^, where its mean and standard deviation were 0.867 and 0.054, respectively. Figure 15c shows the importance levels of the features (IMs, Ns, Nx, and Ny) in which A95, IMcr, AI, and Ic had contributions greater than 0.05.

#### 4.6.2. Second Case Study Using Both the Ground and Roof Sensor Data

Table 13 shows the ML results for the second case, where the IMs are ordered descending from left to right (collected from the feature importance levels greater than 0.05). Even though the difference between R^2^ and the standard deviation for all the ML methods was generally insignificant, the main results are as follows:The maximum R^2^ obtained by the Gradient Boost method is 0.942: R_PGA and R_PGV.The maximum R^2^_mean obtained by the Gradient Boost method is 0.902: R_PGA and R_PGV.The minimum standard deviation obtained by the Linear Regression method is 0.016.The IM present in all the ML methods is R_PGA.

Figure 16 shows the results of the Gradient Boost method of the second case. Figure 16a compares the predicted and reference story drift for the maximum R^2^, which was 0.942. Figure 16b shows the normal distribution function of the R^2^ where its mean and standard deviation were 0.909 and 0.037, respectively. Figure 16c shows the importance levels of the features (IMs, Ns, Nx, and Ny) in which R_PGA and R_PGV had contributions greater than 0.05.

#### 4.6.3. Computation Time

The structural analyses and the ML methodology process were carried out on a computer with 20 Intel^®^ Xeon^®^ W-2255 CPUs @3.70 GHz, 256 Gb of RAM, and 1 NVIDIA RTX A5000 GPU card. The ML algorithms were developed using the Scikit learn library [56] under Python 3.8.3.

The number of structural models was 10,000 per story, considering ten earthquakes, ten scaling factors, ten spans in the X-direction, and ten spans in the Y-direction. Table 14 shows the computation time of the structural analyses per story. The consumed computation time was optimized by running 16 structural models in parallel.

The number of models was 2000 per ML method, considering 10 selections of earthquakes and 200 sets of buildings. Table 15 and Table 16 show the computation time for the first and second cases, respectively.

#### 4.6.4. Discussion of Results

As shown in Figure 15c and Figure 16c, the importance levels of IMs were higher than the structural features of the buildings (Nx, Ny, and Ns). Even though the total number of buildings was 600, the number of record features came from 27 IMs of 10 results (scaling factors) per building. Then, the results depended mainly on the records’ variability, establishing the model’s accuracy. For this reason, it is recommended to increase the number of records in future studies to cover more earthquake features.

Although the accuracy and dispersion for both cases were similar, the main result difference came from the influence of the building response features. Table 12 shows that the main IMs for the first case came from the ground sensors against the second from the roof sensors, as shown in Table 13. Moreover, the main IMs for the first case were based on acceleration and, for the second case, on acceleration and velocity. In addition, for all the ML methods except Decision Tree, the R^2^_mean and the standard deviation were the highest and lowest for the second case, concluding that the second case provides the best high accuracy and low dispersion. However, the inclusion of roof sensors is not easily feasible. Therefore, it is recommended to include roof sensors to increase accuracy and decrease dispersion progressively.

For both cases, Linear Regression and Multilayer Perceptron were the fastest and slowest ML methods, as shown in Table 15 and Table 16. This was because of the complexity of their algorithms, measured by the number of trainable parameters involved. Even though the training and validation could be computationally intensive, once the ML model has been developed it can automatically predict the elastic and inelastic structural responses and detect the damage conditions immediately after the earthquake. For this reason, the Gradient boost (the lowest R^2^_mean) was considered the most effective ML method in both cases in this study.

## 5. Conclusions

This study proposes a methodology to predict the damage conditions of RC resisting-moment frame buildings using ML methods. The methodology was applied to 600 buildings, and the results are summarized as follows:The virtual work method was used to design RC moment-resisting frame system models considering a plastic mechanism, external load, and deformation distribution. The rebar area, distribution of rebars, and realistic member sizes of beams and columns were calibrated using the recommendations of the Japanese standard. The static nonlinear analysis was used to verify the design by comparing the base shear coefficient at the inter-story drift ratio greater or equal to 1/100 with the target value of 0.3.The ground motion records were selected for PGA greater than 400 gals, 5–95% of the Arias intensity time range, and its response spectrum matched the Uniform Hazard Spectrum of Nagoya—Japan (target spectrum) with an exceedance probability of 2% in 50 years.Incremental Dynamic Analyses were carried out on the target buildings in order to obtain the responses covering the linear and nonlinear behavior.Two cases were considered to obtain the Intensity Measures from the sensor records: the first case considered the ground sensors, and the second case considered the ground and roof sensors.Seven machine learning methods were used to predict the damage conditions of the buildings represented by the inter-story drift ratio. The training process used 27 intensity measures obtained from the ground and/or roof sensor responses, the number of stories, and the number of spans in X and Y directions as input data.In order to reduce the bias of the random selection of records and buildings for the training and testing processes, 10 and 200 selections were considered, respectively. An R^2^ mean and standard deviation were obtained for each record selection to evaluate the accuracy of the ML model, and the maximum R^2^ mean and its standard deviation to obtain the best training buildings.For the first case, the maximum R^2^ obtained by the Random Forest method was 0.942, the maximum R^2^ mean obtained by the Gradient Boost method was 0.870, and the minimum standard deviation obtained by the Decision Tree method was 0.047. The IM present in all the ML methods was A95.For the second case, the maximum R^2^ obtained by the Gradient Boost method was 0.942, the maximum R^2^ mean obtained by the Gradient Boost method was 0.902, and the minimum standard deviation obtained by the Linear Regression method was 0.016. The IM present in all the ML methods was R_PGA.The Gradient Boost was considered the most effective ML method in both cases, considering that it has the lowest R^2^_mean.Although the second case presents the highest and lowest R^2^_mean and standard deviation, their inclusion was not easily feasible. It is recommended to include them progressively.It is recommended to increase the number of records in future studies to cover more earthquake features.

Finally, the methodology applied to the RC archetype accurately detected the structural damage condition of the buildings for all ML methods. The Random Forest and Gradient Boosting methods were the most accurate, and the main IMs were those based on acceleration.

## Figures and Tables

**Figure 1 sensors-23-04694-f001:**
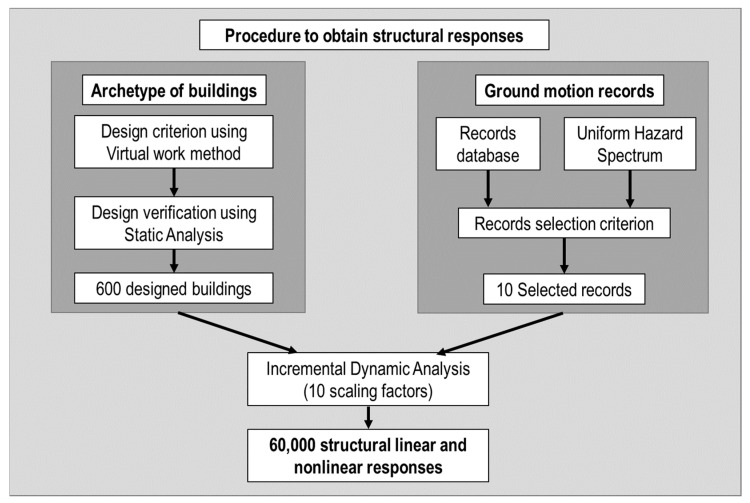
Procedure to obtain structural responses.

**Figure 2 sensors-23-04694-f002:**
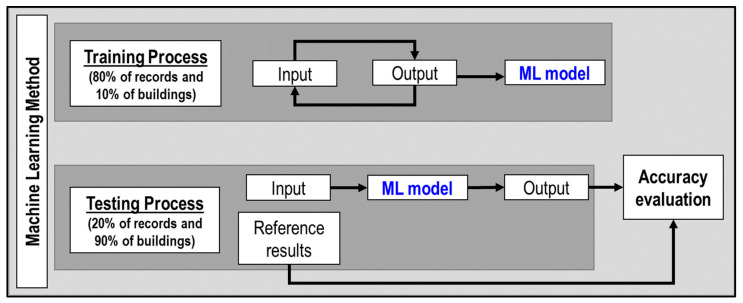
Machine learning method.

**Figure 3 sensors-23-04694-f003:**
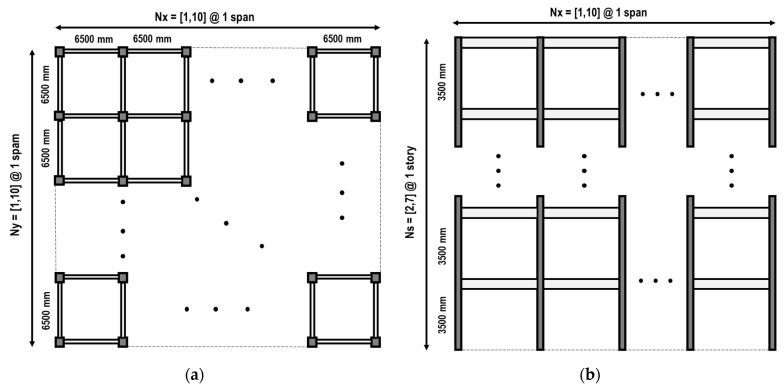
Archetype of buildings: (**a**) plan view; (**b**) elevation view.

**Figure 4 sensors-23-04694-f004:**
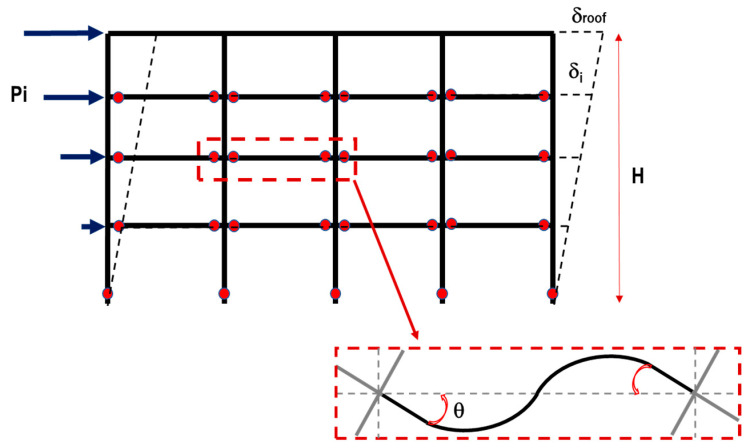
Collapse mechanism and vertical distribution load assumed.

**Figure 5 sensors-23-04694-f005:**
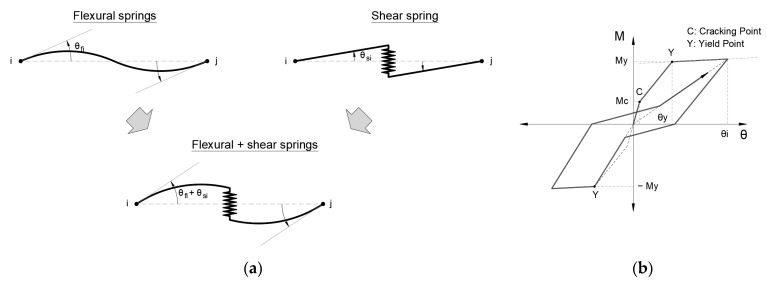
Beam model: (**a**) nonlinear flexural and shear springs; (**b**) degrading trilinear slip hysteresis model [7].

**Figure 6 sensors-23-04694-f006:**
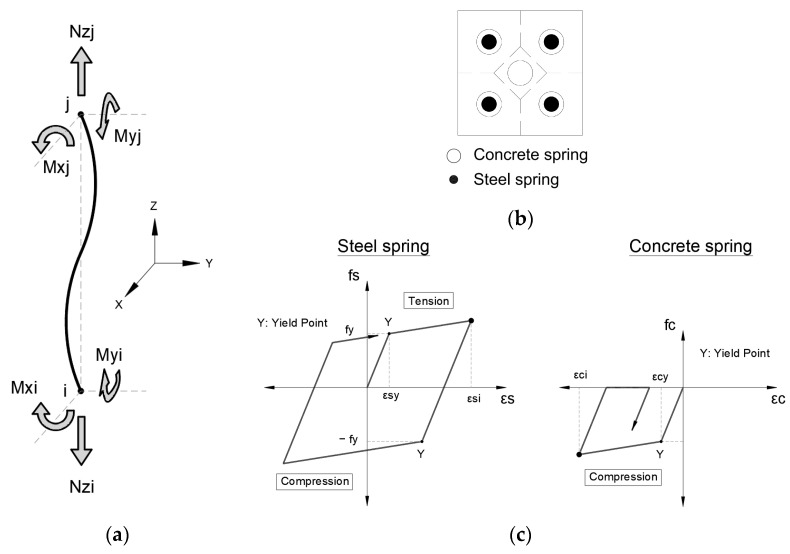
Column model: (**a**) multi-springs to consider Nz-Mx-My nonlinear interaction; (**b**) concrete and steel springs distribution; (**c**) hysteresis model for steel and concrete springs [7].

**Figure 7 sensors-23-04694-f007:**
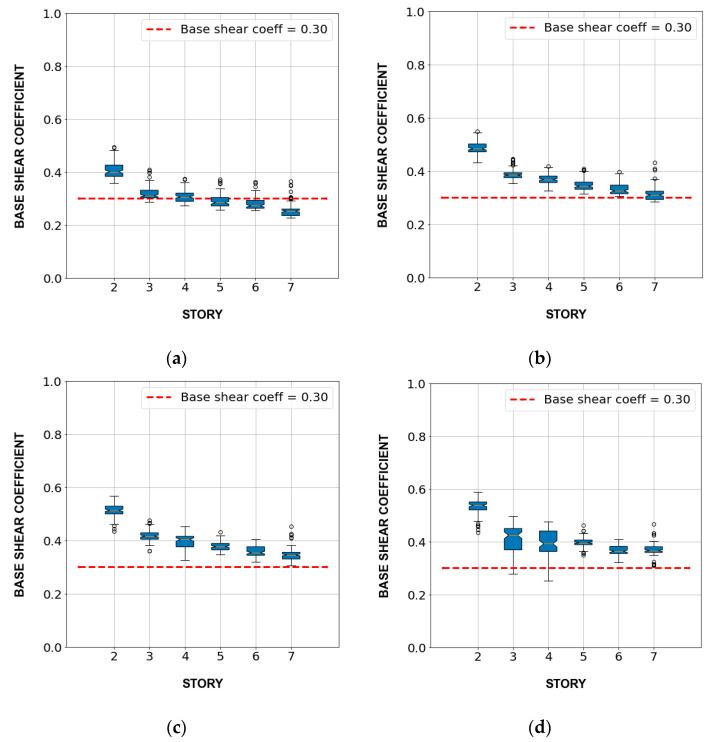
Box plot of base shear coefficient of buildings by the number of stories: (**a**) inter-story drift = 1/150; (**b**) inter-story drift = 1/100; (**c**) inter-story drift = 1/75; (**d**) inter-story drift = 1/50.

**Figure 8 sensors-23-04694-f008:**
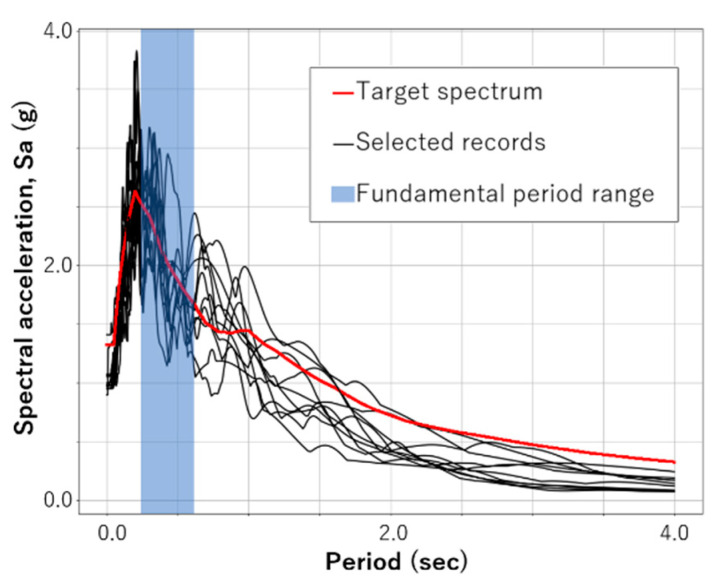
Target spectrum, selected records, and fundamental period range of studied buildings.

**Figure 9 sensors-23-04694-f009:**
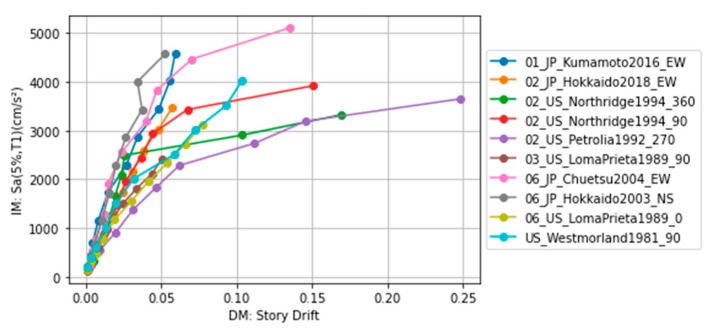
IDA curve for the building of Ns = 3, Nx = 2, and Ny = 5.

**Figure 10 sensors-23-04694-f010:**
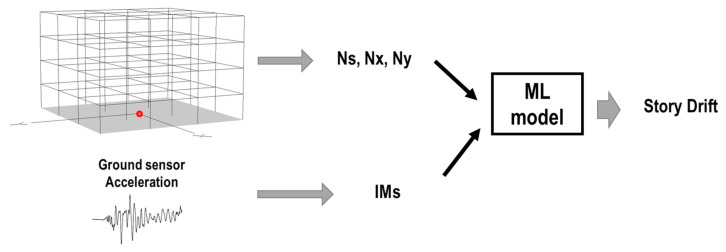
First case: damage detection using only ground sensors of the buildings (red dot represents the location of the sensor).

**Figure 11 sensors-23-04694-f011:**
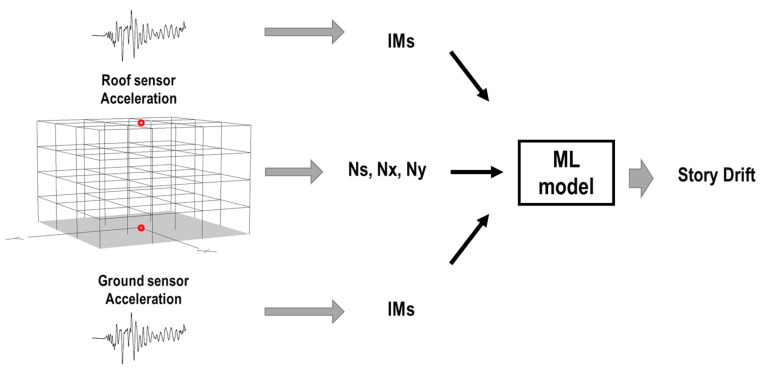
Second case: damage detection using ground and roof sensors of the buildings.

**Figure 12 sensors-23-04694-f012:**
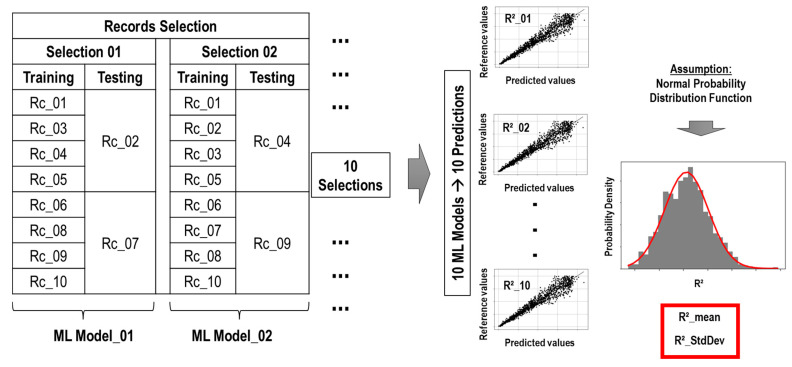
Procedure of the record selection of the ML model.

**Figure 13 sensors-23-04694-f013:**
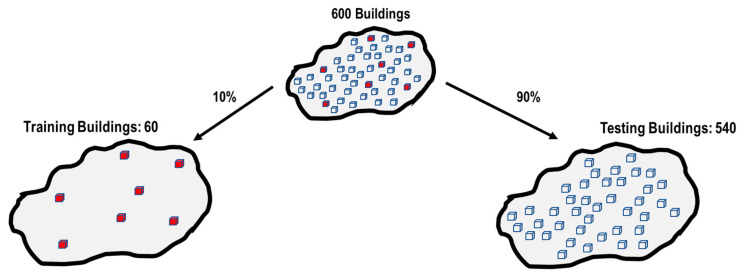
Splitting of buildings.

**Figure 14 sensors-23-04694-f014:**
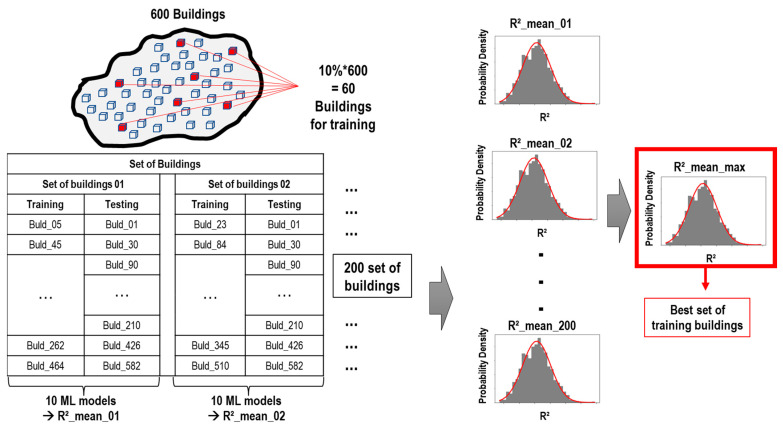
Procedure of the building selection of the ML model.

**Figure 15 sensors-23-04694-f015:**
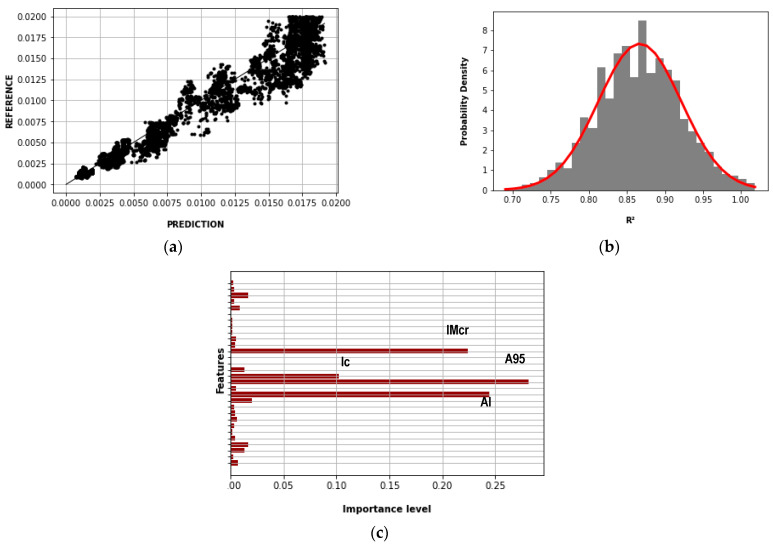
Random Forest results—first case: (**a**) story drift prediction and reference (R^2^ = 0.942); (**b**) normal distribution function of the R^2^ (mean = 0.867; standard deviation = 0.054); (**c**) importance levels of the features.

**Figure 16 sensors-23-04694-f016:**
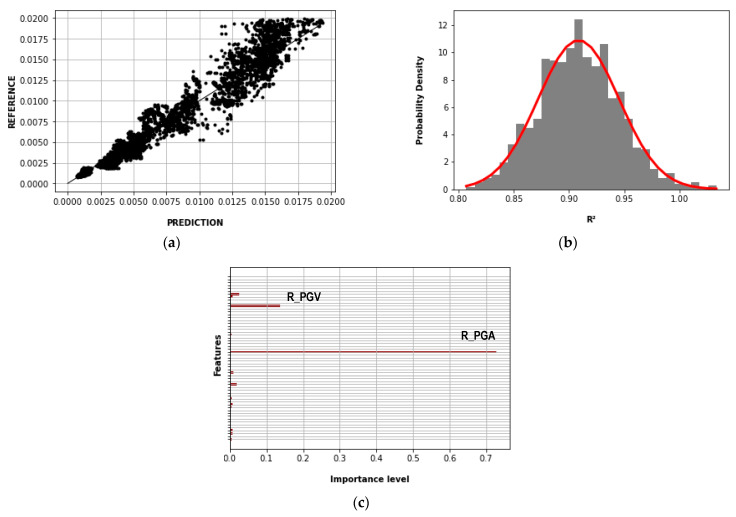
Gradient Boost results—second case: (**a**) story drift prediction and reference (R^2^ = 0.942); (**b**) normal distribution function of the R^2^ (mean = 0.902; standard deviation = 0.037); (**c**) importance levels of the features (IMs).

**Table 1 sensors-23-04694-t001:** The key design variables of the archetype.

Variable	Name	Values
Ns	Number of stories	2 to 7
Nx	Number of spans in the x-direction	1 to 10
Ny	Number of spans in the y-direction	1 to 10
H	Story height [mm]	3500
Lx	Span length [mm]	6500
Ly	Span length [mm]	6500
w	Story weight [kN/m^2^]	10

**Table 2 sensors-23-04694-t002:** Member size of each number of stories.

Number of Stories	2	3	4	5	6	7
Column size [mm]	500 × 500	600 × 600	700 × 700	800 × 800	900 × 900	900 × 900
Beam size (b × $h_{b}$) [mm]	300 × 600	300 × 600	350 × 650	350 × 650	350 × 700	400 × 700

**Table 3 sensors-23-04694-t003:** List of selected records and the scaling factor.

Record Name	$\mathrm{Scaling}\;\mathrm{Factor}\;(SF_{1}$)
Kumamoto2016_EW	1.58
Hokkaido2018_EW	2.24
Northridge1994_360	3.55
Northridge1994_90	3.12
Petrolia1992_270	2.82
LomaPrieta1989_90	2.67
Chuetsu2004_EW	1.36
Hokkaido2003_NS	2.64
LomaPrieta1989_0	2.85
Westmorland1981_90	2.61

**Table 4 sensors-23-04694-t004:** Intensity Measures.

N°	Name	Abbreviation	Based on	Definition	References
1	Peak Ground Acceleration	PGA	A	$PGA=\underset{0\leq t\leq t_{f}}{max}\left|\ddot{u}\right|$	[30]
2	5% Damped First-mode Spectral Acceleration	$S_{a}\left(T_{1},5\%\right)$	A	$S_{a}\left(T_{1},5\%\right)=\left|max({\ddot{u}}_{\left(T_{1},5\%\right)}+{\ddot{u}}_{g})\right|$	[30,31]
3	Average Spectral Acceleration	$Sa_{avg}$	A	Saavg=∏i=1nSaTi1n	[32]
4	Effective Peak Acceleration	EPA	A	$EPA=\frac{1}{2.5}*\int_{0.1}^{0.5}S_{a\left(T,h=5\%\right)}dT$	[33]
5	SR Power-law Form IM	$IM_{SR}$	A	$IM_{SR}=S_{a}{\left(T_{1}\right)}^{1-\alpha }S_{a}{\left(\sqrt{R}T_{1}\right)}^{\alpha }$	[34]
6	CR Power-law Form IM	$IM_{CR}$	A	$IM_{CR}=S_{a}{\left(T_{1}\right)}^{1-\alpha }S_{a}{\left(\sqrt[{3}]{R}T_{1}\right)}^{\alpha }$	[34]
7	Earthquake Power Index	EPI	A	$EPI=\frac{1}{t}*\int_{0}^{t}a_{\left(\tau \right)}{}^{2}d\tau$	[35]
8	Root Mean Square Acc.	RMS	A	$RMS=\sqrt{EPI}$	[35]
9	Bojórquez and Iervolino IM	$I_{N_{P}}$	A	$I_{N_{P}}=S_{a}\left(T_{1},5\%\right)\cdot {\left(\frac{Sa_{avg}}{S_{a}\left(T_{1},5\%\right)}\right)}^{\alpha }$	[36]
10	Arias Intensity	AI	A	$AI=\frac{\pi }{2g}*\int_{0}^{t}a_{\left(\tau \right)}{}^{2}d\tau$	[37]
11	Sarma and Yang IM	$A_{95}$	A	$A_{95}=0.05\cdot \int_{0}^{t}a_{\left(\tau \right)}{}^{2}d\tau$	[38]
12	Characteristic Intensity	$I_{c}$	A	$I_{c}=RMS^{1.5}\cdot t95\_t05^{0.5}$	[39]
13	Riddell and Garcia Acceleration IM	$I_{a}$	A	$I_{a}=a_{max}\cdot t95\_t05^{1/3}$	[40]
14	Cumulative Absolute Velocity	CAV	A	$CAV=\int_{0}^{t}\left|a_{\left(\tau \right)}\right|d\tau$	[41]
15	Standardized Cumulative Absolute Velocity	S−CAV	A	$S-CAV=\overset{N}{\underset{i=1}{\sum }}\left(H_{\left(PGA_{i}-0.025\right)}\int_{i-1}^{i}\left|a_{\left(t\right)}\right|dt\right)$	[42]
16	Two-parameter Hazard IM	TPH	A	RSa=SaTfSaT1TPH=SaT1·RSaα	[43]
17	Peak Ground Velocity	PGV	V	$PGV=\underset{0\leq t\leq t_{f}}{max}\left|v_{\left(t\right)}\right|$	[30,44]
18	Squared Velocity	$V_{sq}$	V	$V_{sq}=\int_{0}^{t}v_{\left(\tau \right)}{}^{2}d\tau$	[19]
19	Root Squared Velocity	$V_{rms}$	V	$V_{rms}=\sqrt{V_{sq}}$	[19]
20	Fajfar et al. IM	$I_{F}$	V	$I_{F}=PGV\cdot t95\_t05^{0.25}$	[45]
21	Riddell and Garcia Velocity IM	$I_{v}$	V	$I_{v}=PGV^{2/3}\cdot t95\_t05^{1/3}$	[40]
22	5% Damped First-mode Spectral Velocity	$S_{v}\left(T_{1},5\%\right)$	V	$S_{v}\left(T_{1},5\%\right)=S_{v\left(T_{1},h\right)}$	[30,31]
23	Housner Spectrum Intensity	$SI_{H}$	V	$SI_{H}=\int_{0.1}^{2.5}S_{V}d\tau$	[46]
24	Peak Ground Displacement	PGD	D	$PGD=\underset{0\leq t\leq t_{f}}{max}\left|u_{\left(t\right)}\right|$	[30]
25	5% Damped First-mode Spectral Displacement	$S_{d}\left(T_{1},5\%\right)$	D	$S_{d}\left(T_{1},5\%\right)=S_{d\left(T_{1},h\right)}$	[30,31]
26	Riddell and Garcia Velocity IM	$I_{d}$	D	$I_{d}=PGD\cdot t95\_t05^{1/3}$	[40]
27	Cosenza and Manfredi IM	$I_{Z}$	H	IZ=∫0tat2dtPGA·PGV	[47]

**Table 5 sensors-23-04694-t005:** Damage condition state [13,14].

Damage Condition	No Damage	Minimum Damage	Significant Damage	Severe Damage	Collapse
Story drift	<1/300	≥1/300 but <1/150	≥1/150 but <1/100	≥1/100 but <1/75	≥1/75

**Table 6 sensors-23-04694-t006:** Decision Tree parameters.

Parameters	Value
Function to measure the quality of a split	MSE
Maximum depth of the tree	No-limit
Minimum number of samples to split	2
Minimum number of leaf nodes	1
Maximum number of leaf nodes	No-limit

**Table 7 sensors-23-04694-t007:** Random Forest parameters.

Parameters	Value
Number of trees in the forest	100
Function to measure the quality of a split	MSE
Maximum depth of the tree	No-limit
Minimum number of samples to split	2
Minimum number of leaf nodes	1
Maximum number of leaf nodes	No-limit

**Table 8 sensors-23-04694-t008:** Gradient Boost parameters.

Parameters	Value
Number of estimators	100
Learning rate	0.1
Function to measure the quality of a split	MSE
Maximum depth of the tree	No-limit
Minimum number of samples to split	2
Minimum number of leaf nodes	1
Maximum number of leaf nodes	No-limit

**Table 9 sensors-23-04694-t009:** AdaBoost parameters.

Parameter	Value
Number of estimators	50
Maximum depth of the tree	3
Minimum number of samples to split	2
Minimum number of leaf nodes	1
Maximum number of leaf nodes	No-limit
Loss function to update the weights	Linear

**Table 10 sensors-23-04694-t010:** XGBoost parameters.

Parameter	Value
Number of estimators	100
Learning rate	0.1
Function to measure the quality of a split	MSE
Maximum depth of the tree	No-limit
Minimum number of samples to split	2
Minimum number of leaf nodes	1
Maximum number of leaf nodes	No-limit

**Table 11 sensors-23-04694-t011:** Multilayer Perceptron parameters.

Parameter	Value
Hidden layer size	100
Maximum number of iterations	100
Learning rate	0.001
Batch size	2
Activation function	ReLU

**Table 12 sensors-23-04694-t012:** The results of the first case study.

Coefficient of Determination (R^2^)							
**Linear Regression**				**Decision Tree**			
**Maximum**	**Mean**	**Standard Deviation**	**Intensity Measure**	**Maximum**	**Mean**	**Standard Deviation**	**Intensity Measure**
0.912	0.820	0.062	-	0.914	0.857	0.047	A95, AI
**Random Forest**				**Gradient Boost**			
**Maximum**	**Mean**	**Standard Deviation**	**Intensity Measure**	**Maximum**	**Mean**	**Standard Deviation**	**Intensity Measure**
0.942	0.867	0.054	A95, IMcr, AI, Ic	0.937	0.870	0.068	A95, AI, Ic, IMcr
**AdaBoost**				**XGBoost**			
**Maximum**	**Mean**	**Standard Deviation**	**Intensity Measure**	**Maximum**	**Mean**	**Standard Deviation**	**Intensity Measure**
0.899	0.857	0.048	SdT1, A95, IMcr, SIH, Ic, IMsr	0.919	0.818	0.089	A95, IF, Sa_Avg, EPA
**Multilayer Perceptron**							
**Maximum**	**Mean**	**Standard Deviation**	**Intensity Measure**				
0.931	0.820	0.065	-				

**Table 13 sensors-23-04694-t013:** The results of the second case study.

Coefficient of Determination (R^2^)							
**Linear Regression**				**Decision Tree**			
**Maximum**	**Mean**	**Standard Deviation**	**Intensity Measure**	**Maximum**	**Mean**	**Standard Deviation**	**Intensity Measure**
0.927	0.897	0.016	-	0.884	0.776	0.075	R_PGA, R_PGV
**Random Forest**				**Gradient Boost**			
**Maximum**	**Mean**	**Standard Deviation**	**Intensity Measure**	**Maximum**	**Mean**	**Standard Deviation**	**Intensity Measure**
0.934	0.893	0.038	R_PGA	0.942	0.902	0.037	R_PGA, R_PGV
**AdaBoost**				**XGBoost**			
**Maximum**	**Mean**	**Standard Deviation**	**Intensity Measure**	**Maximum**	**Mean**	**Standard Deviation**	**Intensity Measure**
0.917	0.896	0.024	R_PGA, R_SIH, R_Sa_Avg, G_Ic, G_CAV	0.93	0.862	0.038	R_PGA, R_PGV, R_IF
**Multilayer Perceptron**							
**Maximum**	**Mean**	**Standard Deviation**	**Intensity Measure**				
0.930	0.881	0.054	-				

**Table 14 sensors-23-04694-t014:** Computation time of the structural analyses per story.

Story	Total Time (10,000 Structural Models per Story) (h)
2	5.95
3	9.82
4	16.37
5	25.20
6	35.83
7	26.68

**Table 15 sensors-23-04694-t015:** Computation time for first case.

ML Method	Training Time per Model (s)	Testing Time per Model (s)	Total Time (2000 Models per ML Method) (s)
Linear Regression	0.0011	0.0008	3.8
Decision Tree	0.0211	0.0010	44.2
Random Forest	1.3454	0.0435	2777.8
Gradient boost	0.5960	0.0047	1201.5
AdaBoost	0.2381	0.0243	524.7
XGboost	0.0929	0.0028	191.5
Multilayer Perceptron	5.3261	0.0153	10,682.9

**Table 16 sensors-23-04694-t016:** Computation time for second case.

ML Method	Training Time per Model (s)	Testing Time per Model (s)	Total Time (2000 Models per ML Method) (s)
Linear Regression	0.0015	0.0010	4.9
Decision Tree	0.0617	0.0015	126.3
Random Forest	3.6086	0.0439	7304.9
Gradient boost	1.8039	0.0074	3622.7
AdaBoost	0.5831	0.0479	1262.0
XGboost	0.1187	0.0029	243.2
Multilayer Perceptron	7.1770	0.0363	14,426.6

## Data Availability

The data presented in this study are available on request from the corresponding author.

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
