# Peer review of "Machine Learning-Based Rapid Post-Earthquake Damage Detection of RC Resisting-Moment Frame Buildings"

_sensors, 2023, doi:10.3390/s23104694_

Round 1

Reviewer 1 Report

The paper presents a study on the prediction of earthquake damages in RC buildings through a ML method. The paper is for sure interesting and it is well written and presented. Nevertheless, some aspect shjould be pointed out before to evalaute the paper suitable for publication. Following my comments: 

- Reading the title and reading the abstract, it seems that title refers to post-earthquake observations while abstract refers to numerical simulations. Thus, I believe that a slight modification should be done in the title, by eliminating the word "post-earthquake"

- The state of the art is poor. I believe that two main topics should be better face: a) The automation in seismic analysis and in modelling, useful for example in large scale analysis; b) the use of ML in civil and structural engineering (see 10.1016/j.engfailanal.2023.107237 and references therein).

- Figure 1 could be improved with some figures. But this is a minor concern. A major observation is that Figures 1 and 2 are not described. 

- What about the record selection? Why did authors select 10 records? Maybe to account for the real source of uncertainty (record-to-record) a good procedure should enlarge the hazard and reduce the varaibility in modelling. 

- Why did authors provide information about IM, when few are well known as good in structural analysis? 

-Regarding IDA, having 3D models, how did authors perform analyses? Take a look to this aspect for 3D models in 10.1002/eqe.3725.

-In the end, how can authors could improve the state of the art with the proposed method? 

Reviewer 2 Report

Overall Decision: Minor revision

In this paper, the machine learning method is proposed to predict the failure conditions of reinforced concrete resistant frame buildings, and the full-text data source uses the virtual working method to design the structural components of 600 reinforced concrete buildings with different floors and spans in the X and Y directions. In order to reduce the error, multiple random selection of buildings and seismic records are used, and seven methods of machine learning are used to train and predict the damage of buildings.The current document had several weaknesses that must be reinforced in order to obtain documentary results equal to the value of publications.

(1) The layout of the paper needs major improvements. First of all, the title of the images throughout the text should be centered, especially the layout of Figure 6, which looks more chaotic; please make changes to clarify the content of the paper. The abscissa axis name of Figure 7 will look neater when centered with the title. It would be better to put the content interpretation of the image in the same interface as the picture, such as Figure 8. Figure 9 is slightly blurry, and a clearer picture should be used. Images of the same type in Figure 15 and Figure 16 should use the same size image format.

(2) Punctuation marks should not appear in formulas, such as Equation (12). The formula number should be on the same level or in the middle line of the formula as the formula.

(3) The table description should also be on the same page and centered as the table content as much as possible, and the same table should be placed on the same page as much as possible. Some tables occupy a large area, and the content layout should be adjusted to make it consistent with the length of the text content. Some tables are drawn inaccurately.

(4) The reference should be indicated by a small mark in the upper right corner. If there are two or more cited papers at the same time, the same document uses the same serial number, and different page numbers are placed after the corresponding serial number in the respective article, for example: XXX[1]204. At the same time, the serial numbers of the references need to be revised.

(5) The document contains a total of 50 employed references, of which 12 are publications produced in the last 5 years (24%), 8 in the last 5-10 years (16%), 29 more than 10 years old (58%), and 1 undated (2%), implying a total percentage of 40% recent references. The proportion of recent references is too small, and the practicality is not high enough; recent references should be added.

(6) The authors may add more state-of-art DL application articles for the integrity of the introduction (An experimental investigation and machine learning-based prediction for seismic performance of steel tubular column filled with recycled aggregate concrete; Reviews on Advanced Materials SciencePrediction of ultimate condition of FRP-confined recycled aggregate concrete using a hybrid boosting model enriched with tabular generative adversarial networks; Thin-Walled Structures.).

(7) It is mentioned that 7 machine learning algorithms are used to make data predictions, and they are compared and placed in a table. However, it is not explained why only the function graph of the random forest algorithm is displayed and whether the training result images of other algorithms are the same or biased. The algorithm section also does not compare which algorithm is more accurate and more suitable for this experiment, only mentioning it in the conclusion section. This is not rigorous enough; please correct it.

(8) Results and Discussion: It is suggested that the author objectively evaluate and analyze this research method based on comparing the research experiments of other scholars. There are also more novel algorithms that can make predictions, and some of this can be added. At the same time, the feasibility and application prospects of the study should also be analyzed.

(9) In the final conclusion, the shortcomings of this paper should be addressed, analyzing what problems and difficulties still exist in the experiment to be solved, and how to further study in the future.

Reviewer 3 Report

The article proposes a machine learning (ML) approach to predict the damage state of reinforced concrete moment frame structures based on seismic loading. The authors obtained sample data by conducting 60,000 time history analyses of 600 different RC structures with varying floor levels and spans in the X and Y directions, using virtual work. Twenty-seven seismic intensity measures (IMs) were employed to capture the structural response, and seven ML methods and two sensor layouts were compared for analysis. While the amount of work presented is commendable, the paper lacks innovation and requires a more in-depth discussion of the results. Specific comments are as follows:

·       The paper's two mentions of scaling factors lack differentiation, which may lead to confusion. Table 3 normalized the response spectra of each wave to match the target spectra, while ten amplitude adjustments were made for each wave during IDA analysis (Figure 9). It is recommended to name the scaling factor used in IDA as SF2 to distinguish it from SF1 in Table 3.

·       The misuse of the term "iteration" in Section 4.4 makes this part extremely difficult to understand. It is easy to misunderstand that the training set and test set were mixed up in this study upon first reading.

·       According to the results in Figure 4.6, the importance levels of IMs are much higher than those of structural features (Nx, Ny, Nz) (Figures 15c and 16c). This suggests that seismic record parameters have a much greater impact on the prediction results than structural parameters, especially when the IM with the highest importance level is A95. However, your training set only includes ten basic seismic records, which is much less than the number of building samples (600). Therefore, I suspect that your training model has poor generalization performance and recommend increasing the seismic record samples.

·       The article lacks innovation in the ML algorithms used, only presenting a simple comparison of different algorithms' prediction performance. However, the paper only shows minor differences in fitting results among the different algorithms, without providing any insightful analyses (such as differences in training time or the factors that cause differences in fitting results). As a result, the paper does not offer valuable insights for future algorithmic improvements. It is recommended to include the authors' interpretations and opinions on this matter.

·       The comparative analysis of the two cases in 4.6 is insufficient, merely listing the results of the two cases without discussing the impact of the two data collection methods on the prediction results. It is recommended to include the authors' interpretations and opinions on this matter.

Round 2

Reviewer 1 Report

After the review provided by authors, the paper can be accepted for publication.

Reviewer 3 Report

The author has replied to all my questions